# Intravenous r-tPA Dose Influence on Outcome after Middle Cerebral Artery Ischemic Stroke Treatment by Mechanical Thrombectomy

**DOI:** 10.3390/medicina56070357

**Published:** 2020-07-17

**Authors:** Marius Kurminas, Andrius Berūkštis, Nerijus Misonis, Karmela Blank, Algirdas Edvardas Tamošiūnas, Dalius Jatužis

**Affiliations:** 1Department of Radiology and Nuclear Medicine, Faculty of Medicine, Vilnius University, Santariškių str. 2, LT-08661 Vilnius, Lithuania; andrius.berukstis@santa.lt (A.B.); nerijus.misonis@santa.lt (N.M.); Algirdas.tamosiunas@mf.vu.lt (A.E.T.); 2Faculty of Medicine, Vilnius University, M. K. Čiurlionio str. 21/27, LT-03101 Vilnius, Lithuania; karmela.blank@santa.lt; 3Centre of Neurology, Faculty of Medicine, Vilnius University, Santariškių str. 2, LT-08661 Vilnius, Lithuania; dalius.jatuzis@mf.vu.lt

**Keywords:** patient outcome assessment, stroke, thrombolysis, thrombectomy, bridging therapy

## Abstract

*Background and Objectives:* Pretreatment with intravenous thrombolysis (IVT) is still recommended in all eligible acute ischemic stroke patients with large-vessel occlusion before mechanical thrombectomy (MTE). However, the added value and safety of bridging therapy versus direct MTE remains controversial. We aimed at evaluating the influence of r-tPA dose level in patients with middle cerebral artery (MCA) occlusion treated with MTE. *Materials and Methods:* We prospectively compared clinical and radiological outcomes in 38 bridging patients, with 65 receiving direct MTE for MCA stroke admitted to Vilnius University Hospital Santaros Clinics. Following our protocol, r-tPA infusion was stopped just before MTE in the operating room. Therefore, we divided all bridging patients into three groups according to the amount of r-tPA they received: bolus, partial dose or full dose. Functional independence at 90 days was assessed by a modified Rankin Scale score, i.e., from 0–2. The safety outcomes included 90-day mortality and any intracerebral hemorrhage (ICH). *Results:* Baseline characteristics and functional outcome at 90 days did not differ between the bridging and direct MTE groups. Shorter MTE procedure and hospitalization time (*p* = 0.025 and *p* = 0.036, respectively) were observed in the direct MTE group. An IVT treatment subgroup analysis showed higher rates of symptomatic ICH (*p* < 0.001) and longer intervals between imaging to MTE (*p* = 0.005) in the full r-tPA dose group. *Conclusions:* In patients with an MCA stroke, direct MTE seems to be a safe and equally effective as bridging therapy. The optimal r-tPA dose remains unclear. Randomized trials are needed to accurately evaluate the added value of r-tPA in patients treated with MTE.

## 1. Introduction

Since the 2015 update of the American Heart Association/American Stroke Association guidelines, the pretreatment with intravenous thrombolysis (IVT) has been recommended in all eligible acute ischemic stroke patients with large-vessel occlusion (LVO) before mechanical thrombectomy (MTE) [1].

However, the added value and safety of pretreatment with IVT before MTE (bridging therapy) in patients considered for MTE remains controversial [2,3]. Historically, in all randomized control trials (RCTs) showing MTE superiority to the best medical treatment (with and without IVT) in anterior circulation ischemic stroke patients with LVO, all IVT-eligible patients received IVT before undergoing randomized MTE or control treatment [4,5,6,7,8]. Still, there are no RCTs to compare the outcomes after complete vs. partial tPA IVT during bridging therapy (BT). Recently released first RCT analyzing bridging therapy results over MTE alone in IVT-eligible patient groups showed no inferiority for dMTE [9]. Two ongoing RCTs (SWIFT DIRECT and MR CLEAN-NO IV) may provide more important variables that may influence the clinical decision to bridge or not. The confrontations regarding the evidence for relative bridging therapy merits over MTE alone exist in recent individual patient and study-level meta-analysis trials [10]. One of them has suggested that BT patients had better functional outcomes, lower mortality, higher rates of successful recanalization and equal odds of symptomatic intracerebral hemorrhage (sICH) compared with patients who received direct MTE [11]. The efficacy of IVT is also questionable, because the bridging therapy itself has no strict regulations. The optimal r-tPA dose in relation to bridging therapy remains unclear, as this type of therapy differs in many centers; for example, not all of them continue the r-tPA infusion during endovascular treatment.

The aim of our prospective observational study was to evaluate the influence of the tissue-type plasminogen activator (r-tPA) dose level in patients treated with MTE for acute ischemic stroke (AIS) related to an occlusion of middle cerebral artery (MCA) by comparing outcomes and complication rates among the patients treated by bridging therapy (BT groups) or direct MTE alone (dMTE group).

## 2. Materials and Methods

In this prospective observational study, we analyzed individual patient data from all consecutive AIS patients treated with MTE for LVO in anterior circulation between February 2015 and August 2018 in Vilnius University Hospital Santaros Clinics. Patients were included if they had MCA occlusion in the M1 segment confirmed by computed tomography angiography (CTA) and were treated with MTE. Sixty-five patients underwent dMTE and thirty-eight received BT. All patients were treated by clinicians as part of their clinical routine; no patients were treated only as part of a study protocol or with the intention to perform research study. The main purpose of our pragmatic study was to clarify whether we should start MTE in cases of LVO only after complete thrombolysis or partial thrombolysis (BT) instead of direct MTE when an operating room was ready immediately. All patients or their legal representatives gave their written informed consent, as previously approved by Vilnius Regional Biomedical Research Ethics Committee (issue number 158200-17-884-407, 10 October 2017).

Baseline characteristics (i.e., demographic data, vascular risk factors), treatment modalities and time from symptom onset to diagnosis and treatment were recorded in the electronic patient case history. Neurological deficit and stroke severity were assessed by certified physicians applying the National Institute of Health Stroke Scale (NIHSS) score. All patients underwent native computed tomography (CT) to exclude intracranial hemorrhage (ICH) and CT angiography with CT perfusion in order to assess the site of vessel occlusion and confirm penumbra ratio. IVT and MTE were performed in compliance with international and institutional guidelines [1,12] (i.e., ischemic stroke or severe head trauma within three months, subarachnoid hemorrhage, recent usage of thrombin inhibitors or factor Xa inhibitors, etc.). Digital subtraction angiography was performed by applying the transfemoral approach using a monoplane, high resolution angiography system. A balloon guided catheter was applied when possible for proximal aspiration, and MTE was carried out with either SOLITAIRE or TREVO stent-retrievers.

The final treatment decision was based on the discretion of the neurologist and neurointerventionalist. If a patient was eligible for IVT, this treatment was started as soon as possible prior to MTE. The IVT dose was calculated according to the patient’s body weight (0.9 mL/kg), and was usually started in the CT scanner room immediately after exclusion of ICH (since 2016) or in the intensive care unit (before 2016). In general, IVT treatment was administered as long as it did not interfere with the patient’s eligibility for treatment with MTE. In some patients, IVT treatment was stopped as soon as the patient was in operating room and the neurointerventionalist was ready to start the MTE procedure. Thus we divided all bridging patients into three groups according to the amount of r-tPA they received: bolus of r-tPA (up to 30% of full dose, BT-B), partial dose of r-tPA (>30% and <75%, BT-p) or full dose of r-tPA (BT-F). If a patient was ineligible for IVT, MTE was started as soon as possible, without pretreatment with IVT. However, in nine cases, when the thrombus length exceeded 8 mm and an operating room was ready, the team decided against BT and instead performed dMTE, even for IVT eligible patients. These decisions were made in compliance with institutional guidelines. Therefore, the dMTE patient group consisted of patients who were IVT eligible and IVT ineligible.

Twenty-four hours after the treatment, or in any case of clinical deterioration, CT was repeated. Symptomatic ICH (sICH) and asymptomatic ICH (aICH) were classified according to the PROACT II (Prolyse in Acute Cerebral Thromboembolism II) study protocol [13]. Recanalization rates were assessed immediately after MTE reperfusion according to the Thrombolysis in Cerebral Infarction (TICI) classification. The clinical outcome was prospectively assessed by a structured telephone interview three months after the stroke using modified Rankin Scale (mRS).

### 2.1. Outcome Measures

The primary study-outcome measure was functional independence at 90 days (assessed by mRS 0–2). Secondary clinical efficacy outcomes were a successful reperfusion following the MTE procedure (defined as a Thrombolysis in Cerebral Infarction (TICI) score of 2b or 3 (complete reperfusion) and a change in NIHSS at 24 h. Safety outcomes included 90-day mortality and any ICH.

### 2.2. Statistical Analysis

Differences between the groups were assessed using the Pearson χ2 test for categorical variables and the Student *t* test or Mann–Whitney U test for continuous variables. The Fisher exact test was applied for continuous variables when the number of observations was less than five. Differences of continuous variables between the four groups were assessed using the nonparametric Kruskal-Wallis test, which was supplemented with the Mann-Whitney U test with Bonferroni correction for analyzing the specific sample pairs for stochastic dominance pairwise. Two-sided *p* values < 0.05 were considered statistically significant. Statistical analysis was performed with SPSS version 25 (IBM Corp., Armonk, NY, USA).

## 3. Results

In the course of the study, 176 patients were treated with MTE. We excluded 73 patients who did not have an occlusion in the M1 segment (Figure 1). One hundred and three patients were divided into groups according to the treatment they received: 38 were treated with bridging therapy intravenous thrombolysis and mechanical thrombectomy, and 65 were treated with direct mechanical thrombectomy (dMTE) alone. As explained above, patients in BT received various percentages of the calculated IVT dose before MTE.

### 3.1. BT vs. dMTE

There were no statistically significant differences in baseline characteristics between the two groups except for higher rates of use of anticoagulants prior to AIS in the dMTE group (*p* = 0.022) (Table 1). The median duration time of MTE and the duration of hospitalization were longer in the BT group (*p* = 0.025 and *p* = 0.036, respectively). Functional independence (*p* = 0.814) (Figure 2), successful reperfusion (*p* = 0.717), NIHSS change during first 24 h (*p* = 0.665), mortality at 90 days (*p* = 0.638) and sICH (*p* = 0.121) did not differ significantly between the two groups. During the study period, we did not observe complete recanalization cases in the BT group. Only one from the excluded (M2) group showed no occlusions in the operating room; however, three of the patients from our target (M1) group showed partial lysis/thrombus migration into M2 segments.

### 3.2. Various Dosages of r-tPA vs. dMTE

There was a significant difference among the groups regarding the median time from the first contact with the neurologist to MTE (*p* = 0.018), and the median time from image to MTE (*p* = 0.005). Functional independence (*p* = 0.427) (Figure 3), successful reperfusion (*p* = 0.825), NIHSS change during the first 24 h (*p* = 0.990) and mortality at 90 days due to AIS (*p* = 0.905) did not differ among the groups. However, there was a significant difference in sICH (*p* < 0.001) (Table 2).

We performed statistical analyses of different group pairs for those parameters that had significant differences. There was a significant difference of sICH between the dMTE and BT-F groups (*p* < 0.001), the median time from image to MTE and the median time from the neurologist’s consultation to MTE between the dMTE and BT-F groups (*p* = 0.005 and *p* = 0.010, respectively), between the BT-B and BT-F groups (*p* = 0.006 and *p* = 0.020, respectively) and between the BT-p and BT-F groups (*p* = 0.001 and *p* = 0.025, respectively) (Table 3).

## 4. Discussion

Our study showed that, in our small stroke center environment, the increasead sICH rate was present in MCA stroke pateints treated with for full dose BT. Additionally, and that IVT delayed the initiation of MTE for this group of patients. Recent metaanalyses provided nonsignificant evidence that IVT added to MTE increases the likelihood of sICH (sOR 0.96, 95% CI 0.63 to 1.17) [14]. In the majority of the studies, there was no difference in terms of procedural complications, despite thrombolysis-induced coagulopathy [15], and only one matched-paired analysis showed an increased risk of asymptomatic intracranial hemorrhage [16]. Regarding the heterogeneity of patient groups, in most studies, they were not directly comparable because IVT-ineligible (dMTE) patients usually have more comorbidities; in these studies, subjects were not randomized to receive IVT. Delayed presentation and the use of anticoagulants were the most common reasons for IVT ineligibility [11]. Only a few studies have assessed the safety and efficacy of dMTE in IVT-eligible patients vs. bridging therapy [16,17,18]. This may explain the higher mortality rate in the dMTE group [15]. The most recent data from the DIRECT-MT study showed that endovascular thrombectomy alone was noninferior with regard to functional outcome to endovascular thrombectomy preceded by intravenous alteplase administered within 4.5 h after symptom onset. However, the vast majority of patients in the combination-therapy group in this study (93.7%) received a full dose of tPA, which precluded the comparison of outcomes after complete vs. partial tPA IVT during BT [9]. Even if the homogeneity of the groups regarding the IVT eligibility could solve the mortality issue, there is still no new information regarding the merits of BT over dMT; only 23 out of 319 patients completed the infusion before groin puncture. Bleeding complications, including numerically higher sICH in the combination-therapy group, keep the BT question open: is it worth continuing tPA infusion during and even after completion of the thrombectomy? In our study, patients who were eligible for IVT did not necessarily receive r-tPA according to the decision of the neurologist and the neurointerventionalist team. No differences in the basic patient characteristics were found (Table 1).

To date, there are some grey areas regarding bridging therapy; therefore, in some studies, not all patients received a full dose of r-tPA before MTE [17,19]. Many centers continue r-tPA infusion during MTE, and some do not stop thrombolysis even after successful recanalization achieved by MTE according to the presumption that rt-PA may help to dissolve very distal thrombi [14,20,21]. We stopped the infusion just before the arterial puncture in order to avoid potential complications. That could probably explain our significant differences in numbers, showing MTE initiation delay for this group of patients (door/imaging—MTE time). Therefore, the total treatment time (onset to MTE or to reperfusion) showed no significant differences. In recently published data [2], like in our study, IVT was shown to delay the initiation of MTE. Mostly this is related to the drip and ship approach [22]; however, our data showed that the time from first neurologist visit to MTE and time from image to MTE were significantly longer in the BT-F group in comparison to other groups. In our full dose r-tPA group, there were three patients from regional hospitals who received no ICH after treatment.

To minimize the bias of our nonrandomized trial, we included only MCA occlusions. This is a classical indicator of anterior circulation stroke, usually with clear symptoms. We believe that this segment is the easiest to compare according to the clinical and pathophysiological presentation, and to the functional and morphological outcomes. We discovered neither tendencies for higher rates of early neurological improvement nor favorable outcomes within three months in IVT pretreated patients, like Guedin et al. in their study on a population of 68 patients with MCA occlusion [23]. The factor which had a major influence on the outcome [24] was the rate of successful recanalization, which was similar in all groups (*p* = 0.890). During the study period, only three of the patients from our group with clot lengths slightly below the average (5, 12 and 14 mm) showed partial lysis-thrombus migration into M2 segments. This result does not contradict the calculations by Behrens et al., who showed that no thrombus longer than 16mm could be completely recanalized by tPA [25]. Bhogal et al. suggest consideration of dMTE for all patients in whom the clot is 4 mm or longer without bridging [26].

The main limitations were the monocentric nonrandomized study design and the small and uneven sample size, especially in the BT groups. This should be taken in consideration when looking for statistical differences between the outcomes of subgroups. Multinomial regression analysis was not performed because the small sample size of cases after accounting for multicollinearity and outliers would be too small to make any statistical sense.

## 5. Conclusions

The value of IVT for patients with LVO, treated with MTE, is one of the most controversial topics of acute stroke care. Our results show a tendency towards higher risk of sICH rate for patients with MCA occlusion who have received a full dose of r-tPA before MTE. In the overall, low-quality evidence regarding the relative bridging therapy merits, our findings highlight the need for RCTs.

## Figures and Tables

**Figure 1 medicina-56-00357-f001:**
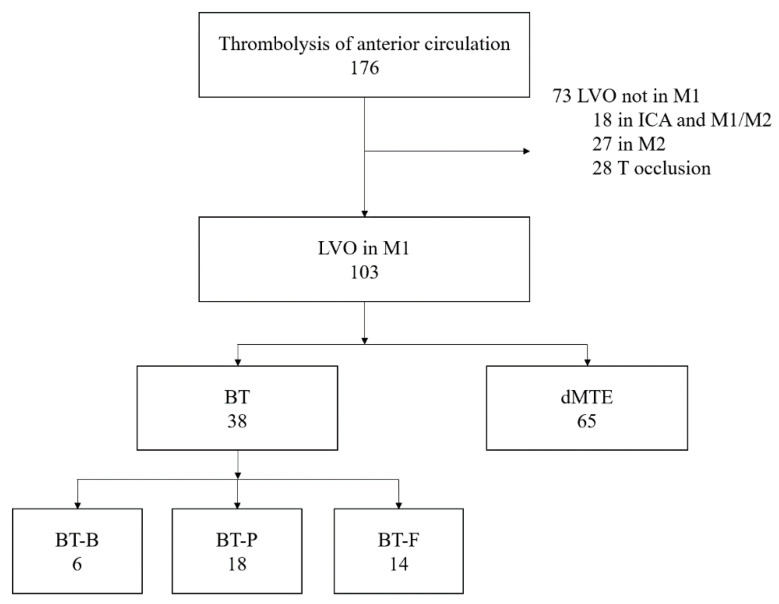
Patient selection and division into groups. Patients were initially selected from a separate anterior circulation thrombolysis log. Patients were excluded if they had large vessel obstruction (LVO) not in the M1 segment of middle cerebral artery (MCA), any combination of several LVO, or T type occlusions. Patients were divided into two large groups according to the received treatment strategy: bridging therapy (BT) group—intravenous thrombolysis (IVT) and mechanical thrombectomy (MTE); dMTE—direct mechanical thrombectomy group. The BT group was divided into three subgroups according to the actual r-tPA dose received during IVT (BT-B—bolus of r-tPA (up to 30% of full r-tPA dose) followed by MTE; BT-P—partial dose of r-tPA (>30% and <75%) followed by MTE; BT-F—full dose of r-tPA followed by MTE). ICA—internal carotid artery.

**Figure 2 medicina-56-00357-f002:**
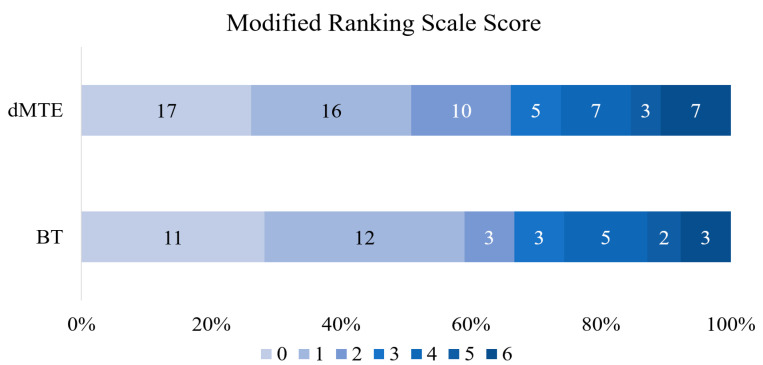
Distribution of modified Rankin scale in two groups after 3 months. BT—bridging therapy; dMTE—direct mechanical thrombectomy.

**Figure 3 medicina-56-00357-f003:**
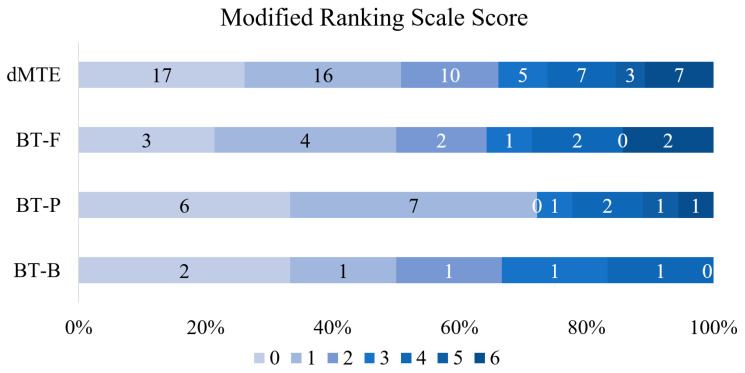
Distribution of modified Rankin scale in four groups after 3 months. BT-B—bridging therapy with bolus dose of intravenous thrombolysis; BT-F—bridging therapy with full dose of intravenous thrombolysis; BT-P—bridging therapy with partial dose of intravenous thrombolysis; dMTE—direct mechanical thrombectomy.

**Table 1 medicina-56-00357-t001:** Baseline characteristics and outcomes in two groups analysis.

	BT (n = 38)	dMTE (n = 65)	*p* Value
Baseline characteristics			
Sex female, n (%)	22 (57.9)	39 (60)	0.835
Age, y mean (SD)	67.1 (9.6)	68.4 (11.8)	0.347
NIHSS on admission, median (range)	14 (4–20)	13 (2–24)	0.635
ASPECTS on admission CT, median (range)	9 (6–10)	9 (6–10)	0.689
Clot length mm, mean (SD)	14.8 (8.2)	15.7 (7.1)	0.452
Vascular and other risk factors, n (%)			
Hypertension	34 (89.5)	53 (81.5)	0.286
Diabetes mellitus	4 (10.5)	14 (21.5)	0.158
Atrial fibrillation	21 (58.3)	33 (50.8)	0.468
Coronary heart disease	17 (44.7)	32 (49.2)	0.661
Heart failure	15 (39.5)	23 (35.4)	0.680
Use of anticoagulants, n (%)	3 * (7.89)	17 (28.33)	0.022
Vital signs			
Systolic blood pressure, mean (SD)	150.7 (25.4)	153.9 (28.2)	0.983
Diastolic blood pressure, mean (SD)	85.0 (12.3)	85.7 (12.3)	0.321
Mean blood pressure, mean (SD)	109.8 (14.4)	108.5 (17.8)	0.485
Pulse, mean (SD)	80.2 (14.9)	82.3 (23.8)	0.524
Treatment			
Median time from neurologist’s consultation to MTE, h (IQR)	01:23:00 (00:30:45)	01:12:00 (00:27:30)	0.113
Median time from image to MTE, h (IQR)	00:46:30 (00:37:45)	00:42:30 (00:26:15)	0.625
Median time from symptom onset to recanalization, h (IQR)	04:00:00 (01:12:30)	03:50:30 (02:07:30)	0.865
Median time from symptom onset to MTE, h (IQR)	03:10:00 (01:14:00	03:00:00 (02:19:00)	0.994
Median duration of MTE, h (IQR)	00:42:30 (00:30:00)	00:30:00 (00:25:00)	0.025
Duration of hospitalization, days, mean (SD)	24.5 (4.4)	21.4 (2.6)	0.036
Outcome			
Successful reperfusion (TICI 2b-3), n (%)	33 (86.8)	58 (89.2)	0.717
Complete reperfusion (TICI 3), n (%)	22 9 (57.9)	35 (53.9)	0.691
NIHSS change during first 24 h, median (range)	5 (−5, 17)	6 (−6, 19)	0.665
Complications, n (%)			
sICH	4 (10.5)	1 (1.5)	0.121
aICH	4 (10.5)	7 (10.8)	0.831
Distal embolization	5 (13.2)	8 (12.3)	0.787
Clinical outcome after 3 months			
Functional independence (mRS 0–2), n (%)	26 (68.4)	43 (66.2)	0.814
Excellent clinical outcome (mRS 0–1), n (%)	23 (60.5)	33 (50.8)	0.340
Mortality at 90 days, n (%)	2 (5.3)	5 (7.7)	0.638

aICH—asymptomatic intracerebral hemorrhage, ASPECTS—Alberta Stroke Program Early CT Score, BT—bridging therapy, CT—computed tomography, dMTE—direct mechanical thrombectomy, mRS—Modified Rankin Score, MTE—mechanical thrombectomy, NIHSS—National Institute of Health Stroke Score, sICH—symptomatic intracerebral hemorrhage, TICI—Thrombolysis in Cerebral Infarction score. * INR < 1.11.

**Table 2 medicina-56-00357-t002:** Baseline characteristics and outcomes in four group analysis.

	BT-B (n = 6)	BT-*P* (n = 18)	BT-F (n = 14)	dMTE (n = 65)	*p* Value
Baseline characteristics					
Sex female, n (%)	3 (50.0)	12 (66.7)	7 (50.0)	39 (60.0)	0.722
Age, y mean (SD)	65.3 (7.5)	67.6 (10.3)	67.4 (9.3)	68.4 (11.8)	0.649
NIHSS on admission, median (range)	9.5 (5–18)	14 (4–20)	13.5 (5–20)	13 (2–24)	0.581
ASPECTS on admission CT, median (range)	9 (7–10)	9.5 (7–10)	8.5 (6–10)	9 (6–10)	0.329
Clot length mm, mean (SD)	16.2 (8.3)	13.6 (5)	18.3 (8.4)	14.8 (8.2)	0.360
Vascular and other risk factors, n (%)					
Hypertension	5 (83.3)	16 (88.9)	13 (92.9)	53 (81.5)	0.697
Diabetes mellitus	1 (16.7)	1 (5.6)	2 (14.3)	14 (21.5)	0.459
Atrial fibrillation	2 (33.3)	11 (61.1)	8 (57.1)	33 (50.8)	0.729
Coronary heart disease	1 (16.7)	9 (50.0)	7 (50.0)	32 (49.2	0.490
Heart failure	2 (33.3)	7 (38.9)	6 (42.9)	23 (35.4)	0.953
Use of anticoagulants, n (%)	0 (0.0)	2 * (11.1)	1 * (7.1)	17 (28.3)	0.134
Vital signs					
Systolic blood pressure, mean (SD)	146.0 (25.1)	158.1 (28.7)	153.7 (21.7)	153.9 (28.2)	0.711
Diastolic blood pressure, mean (SD)	82.6 (4.3)	86.0 (13.4)	90.6 (12.4)	85.7 (12.3)	0.449
Mean blood pressure, mean (SD)	103.7 (11.2)	110.0 (16.3)	111.6 (13.1)	108.5 (17.8)	0.608
Pulse, mean (SD)	71.6 (6.3)	83.9 (19.0)	78.3 (8.2)	82.3 (23.8)	0.402
Treatment					
Median time from neurologist’s consultation to MTE, h (IQR)	00:59:00 (00:31:00)	01:21:00 (00:27:00)	01:34:00 (00:45:00)	01:12:00 (00:27:30)	0.017
Median time from image to MTE, h (IQR)	00:28:00 (00:24:00)	00:38:00 (00:20:00)	01:13:00 (00:39:00)	00:42:30 (00:26:15)	0.005
Median time from symptom onset to MTE, h (IQR)	02:24:30 (01:50:00)	03:05:00 (01:03:00)	03:30:00 (01:17:00)	03:00:00 (02:15:00)	0.792
Median time from symptom onset to r-tPA, h (IQR)	01:57:00 (02:10:15)	02:22:30 (01:26:15)	01:49:00 (00:53:30)		0.280
Median time from symptom onset to recanalization, h (IQR)	03:22:00 (01:47:00)	03:55:00 (01:04:00)	04:12:00 (01:33:00)	03:50:30 (02:07:30)	0.531
Median duration of MTE, h (IQR)	00:50:00 (00:32:00)	00:40:00 (00:35:00)	00:45:00 (00:30:00)	00:30:00 (00:25:00)	0.125
Duration of hospitalization (days), mean (SD)	28.3 (14.6)	21.3 (5.5)	27.0 (7.9)	21.4 (2.6)	0.928
Outcome					
Successful reperfusion (TICI 2b-3), n (%)	5 (83.3)	15 (83.3)	13 (92.9)	58 (89.2)	0.825
Complete reperfusion (TICI 3), n (%)	3 (50.0)	11 (61.1)	8 (57.1)	35 (53.8)	0.943
NIHSS change during first 24 h, median (range)	6 (1–13)	6.8 (−5, 17)	6 (−5, 16)	6 (−6, 19)	0.990
Complications, n (%)					
sICH	0 (0)	0 (0)	4 (28.6)	1 (1.5)	<0.001
aICH	0 (0)	1 (5.6)	3 (21.4)	7 (10.8)	0.410
Distal embolization	1 (16.7)	3 (16.7)	1 (7.1)	8 (12.3)	0.865
Clinical outcome after 3 months					
Functional independence (mRS 0–2), n (%)	4 (66.7)	13 (72.2)	9 (64.3)	43 (66.2)	0.427
Excellent clinical outcome (mRS 0–1), n (%)	3 (50)	13 (72.2)	7 (50)	33 (50.8)	0.926
Mortality at 90 days, n (%)	0 (0)	1 (5.6)	1 (7.1)	5 (7.7)	0.905

aICH—symptomatic intracerebral hemorrhage, ASPECTS—Alberta Stroke Program Early CT Score, BT-B—bridging therapy with bolus dose of intravenous thrombolysis, BT-F—bridging therapy with full dose of intravenous thrombolysis, BT-P—bridging therapy with partial dose of intravenous thrombolysis, CT—computed tomography, dMTE—direct mechanical thrombectomy, mRS—modified Rankin score, MTE—mechanical thrombectomy, NIHSS—National Institute of Health Stroke Score, sICH—symptomatic intracerebral hemorrhage, TICI—Thrombolysis in Cerebral Infarction score. * INR < 1.11.

**Table 3 medicina-56-00357-t003:** Pair comparison of statistically significant parameters.

	Groups Pairs	dMTE and BT-B	dMTE and BT-P	dMTE and BT-F	BT-B and BT-P	BT-B and BT-F	BT-P and BT-F
Dependent Variable							
sICH		1.000	1.000	<0.001	1.000	0.040	0.001
Median time from neurologist’s consultation to MTE		1.000	1.000	0.036	0.719	0.041	0.597
Median time from image to MTE		0.932	1.000	0.029	1.000	0.019	0.015

BT-B—bridging therapy with bolus dose of intravenous thrombolysis, BT-F—bridging therapy with full dose of intravenous thrombolysis, BT-P—bridging therapy with partial dose of intravenous thrombolysis, dMTE—direct mechanical thrombectomy, MTE—mechanical thrombectomy, sICH—symptomatic intracerebral hemorrhage.

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
