# Peer review of "Intravenous r-tPA Dose Influence on Outcome after Middle Cerebral Artery Ischemic Stroke Treatment by Mechanical Thrombectomy"

_medicina, 2020, doi:10.3390/medicina56070357_

Round 1
Reviewer 1 Report
No further comments.
Author Response
Thank you for the positive review.Reviewer 2 Report
Thank you for your submission and interesting review of cases.
I have the following comments.
In your introduction you should mention DIRECT-MT as this was recently published and does explore the rtPA vs. no rtPA prior to MT debate. You stipulate that:
Still there are no RCTs to compare 42 bridging therapy results over MTE alone, this only will be assessed in two ongoing RCTs (SWIFT DIRECT and MR CLEAN-NO IV).
I think it would be useful to inlcude a slightly more expansive discussion on the failings of previous studies looking at this topic e.g. self slecting of worse cases for direct to MT because they are already out of time window for tpa and therefore likely large stroke volume etc.
You mention that when the clot was 8mm or longer you went direct to MT. WHilst I agree with this and realise there is little evidence in favour of doing this this concept is actually wrong. The correct concept should be - at what length of clot does the chance of revascularisation with MT exceed that of recanalisation with IV rtPA - this cross over of possibilities is the important thing and will occur at short clot lengths. This concept was discussed by Bhogal at el in their paper - by their rough calculations MT should be offered where clot length is 4mm:
Mechanical Thrombectomy in Patients With M1 Occlusion and NIHSS Score ≤5: A Single-Centre Experience
It would be useful to add the length + range of the clots in your study and then compare the usefulness of the different treatment strategirs dMTE, TPA-B, TPA-F etc etc I think the discussion can be expanded with a critique of DIRECT MT study to highlight some of the flaws in that study.Author Response
Response to Reviewer 2 Comments
Thank you for your submission and interesting review of cases
Response: thank you for the positive comment and encouragement.
Point 1: In your introduction you should mention DIRECT-MT as this was recently published and does explore the rtPA vs. no rtPA prior to MT debate. You stipulate that:
Still there are no RCTs to compare 42 bridging therapy results over MTE alone, this only will be assessed in two ongoing RCTs (SWIFT DIRECT and MR CLEAN-NO IV).
Response: thank you for this important point. We revised the text to clarify the Introduction section (lines 43-47):
Still there are no RCTs to compare the outcomes after complete vs partial tPA IVT during BT, just very recently released first RCT analyzing BT results over MTE alone in IVT-eligible patient groups showed no inferiority for dMTE [9] and maybe two ongoing RCTs (SWIFT DIRECT and MR CLEAN-NO IV) will provide more important variables that may influence the clinical decision to bridge or not to bridge. The confrontations regarding the evidence for relative bridging therapy merits over MTE alone exist in recent individual patient and study-level meta-analysis trials [10].
Point 2: I think it would be useful to include a slightly more expansive discussion on the failings of previous studies looking at this topic e.g. self selecting of worse cases for direct to MT because they are already out of time window for tpa and therefore likely large stroke volume etc.
Response: We clarified the first paragraph in the discussion section (lines 200-201):
Regarding the heterogeneity of patients groups, in most of the studies they are not directly comparable, because IVT–ineligible (dMTE) patients have usually more comorbidities, in these studies they were not randomized to receive IVT. Delayed presentation and use of anticoagulants were the most common reasons for IVT ineligibility [11]. Only few studies have assessed the safety and efficacy of dMTE in IVT-eligible patients vs bridging therapy [16–18]. This may explain higher mortality in the dMTE group [15].
Point 3: You mention that when the clot was 8mm or longer you went direct to MT. Whilst I agree with this and realise there is little evidence in favour of doing this, this concept is actually wrong. The correct concept should be - at what length of clot does the chance of revascularisation with MT exceed that of recanalisation with IV rtPA - this cross over of possibilities is the important thing and will occur at short clot lengths. This concept was discussed by Bhogal at el in their paper - by their rough calculations MT should be offered where clot length is 4 mm:
Mechanical Thrombectomy in Patients With M1 Occlusion and NIHSS Score ≤5: A Single-Centre Experience
P Bhogal 1 , P Bücke 2 , O Ganslandt 3 , H Bäzner 2 , H Henkes 1 4 , M Aguilar Pérez 1
It would be useful to add the length + range of the clots in your study and then compare the usefulness of the different treatment strategirs dMTE, TPA-B, TPA-F etc etc
Response: Thank you for raising this important point. We totally agree that still there is no answer at what length of clot does the chance of revascularization with MTE exceed that of recanalization with IV tPA for a specific patient. We did our calculations and there was no difference between the groups regarding the length and range of the clots. We did not observed the clot length (different < 8; 16 mm groups) influence on our study outcomes.
We added the clot length data in Tables 1 and 2. Additionally we expanded our discussion section (lines 245-250):
The factor which has a major influence on the outcome [24] is the rate of successful recanalization, it was also similar in all groups (p=0.890). During the study period only three of the patients from our group with clot length slightly below the average (5, 12 and 14 mm) have showed partial lysis – thrombus migration into M2 segments. This result does not contradict with the calculations by Behrens et al in their study showing that no thrombus longer than 16 mm could be completely recanalized by tPA [25]. Bhogal et all suggest to consider dMTE for all patients in whom the clot is 4 mm or longer without bridging [26].
Point 4: I think the discussion can be expanded with a critique of DIRECT MT study to highlight some of the flaws in that study.
Response: thank you for this suggestion. We expanded the discussion section (lines 208-213) accordingly.
The most recent data from the DIRECT-MT study showed that endovascular thrombectomy alone was noninferior with regard to functional outcome to endovascular thrombectomy preceded by intravenous alteplase administered within 4.5 hours after symptom onset. However, the vast majority of the patients in this study (93.7%) in the combination-therapy group received full dose of tPA, which precludes a comparison of outcomes after complete vs partial tPA IVT during BT [9]. Even if the homogeneiety of the groupsregarding the IVT eligibility could solve the mortality point, still there is no answer regarding the BT merits over dMTE, only 23 of 319 patients completed the infusion before groin puncture. The bleeding complications, including numerically higher sICH in combination-therapy group keeps BT question open – does it worth to continue tPA infusion during and even after the completion of thrombectomy?
Round 2
Reviewer 2 Report
The authors have done a good job of amending the article. I have no further comments.
This manuscript is a resubmission of an earlier submission. The following is a list of the peer review reports and author responses from that submission.
Round 1
Reviewer 1 Report
A non-randomized comparison of outcomes in stroke patients undergoing mechanical thrombectomy (MTE) treated with (bridging) or without (direct) tPA. There is equipoise in this area because tPA works less well against large clots so that the benefit may not be worth the risk and potential delay. Well-designed randomized trials are the only way to resolve this question, but in the meantime, there have been a proliferation of small case series, of which this is an example. This series has two unique features distinguishing it from others. Bridging patients did not all receive the complete infusion so that patients receiving only bolus, partial or full dose were analyzed separately. Also, patients ineligible for tPA were included in the direct group. Results were no better in the bridging group with more delay to MTE and higher rates of symptomatic hemorrhage.
Despite (and in part because of) its unique features, the study has numerous deficiencies that limit the credibility of the results. They are (in no particular order of importance)…….
- The sample sizes of each of the 3 bridging subgroups are so small that it is highly surprising that any differences were statistically significant. It does not appear that the authors used multivariable regression in comparing outcomes between the subgroups.
- In the bridging group, what was the time from symptom onset or last known well to bolus? The likelihood of tPA benefit is in the first two hours from onset so if such patients were not included, it could explain the lack of benefit.
- Further, recanalization of LVOs after tPA, when it happens, usually doesn’t occur until towards the end or following the completion of the infusion. Not allowing the complete infusion will bias the results against the bridging group.
- 3 patients in the bridging group used anticoagulants. Therefore, these patients were not “tPA eligible” and may have inflated the risk of subsequent hemorrhage.
- Of course, the bridging and direct groups were not randomized, and even though baseline demographics were similar, many other non-reported variables may have contributed to the decision to bridge or not bridge.
- Along that line, if the authors want to compare bridging to non-bridging, then only tPA eligible patients should be included.
- Going further, the fundamental problem with this analysis, and with all such analyses to date, is that they are posited from the perspective of the clinician carrying out MTE, and are not relevant to the clinician who has to decide in the ED whether to give tPA or not to a potential MTE candidate. When these patients are seen, often in a peripheral hospital, or even in the hub ED, it is never certain that they will go to MTE or if so, how long will be the delay. So the only way to answer the question of whether patients should be bridged is to randomize the patients in an intention to treat fashion at the time that decision has to be made. This means including in the analysis the patients who receive tPA, recanalize or substantially improve before getting to the MTE suite, and therefore never have MTE. Focusing only on patients presenting to the MTE center who need MTE “harvests out” all the patients who have benefitted from bridging.
Reviewer 2 Report
The authors conducted a prospective, observational not randomized comparison between patients with AIS and LVO treated with IV-tPA plus MTE (bridging) (n=38) or direct MTE (65) in order to evaluate the added value and safety of pretreatment of tPA on MTE.
The study was well designed and conducted and the results are important and clinically relevant. The paper is well written but there are several points that should be addressed.
- In the Material and Methods:
- The authors should explain why only M1 occlusions were included in the analysis. It will be interesting to know how many ICA or T occlusions MTE were performed wither by dMTE or BT
- Were all the NIHSS assessments done by certified physicians?
- Were all the MTE on LVOs performed within 4.5 hours of symptom onset? If so, what were the contradictions to tPA?
- Was the mRS assessed at 90 days performed by an assessor blinded to the intervention? This is in order to avoid bias.
- What was the total number of AIS admissions during the study period
- What was the percentage of tPA treatments in the total AIS group? These 2 points describe the magnitude of the stroke center.
- In the Results section - How many of the patients in the BT have shown complete recanalization at the OR before MT had started?
In the Discussion- a recent Chinese study was published in the NEJM (May 6, 2020) Direct MT by Jiannin Liu et.al and should be added to the Discussion